# Machine Learning System for Lung Neoplasms Distinguished Based on Scleral Data

**DOI:** 10.3390/diagnostics13040648

**Published:** 2023-02-09

**Authors:** Qin Huang, Wenqi Lv, Zhanping Zhou, Shuting Tan, Xue Lin, Zihao Bo, Rongxin Fu, Xiangyu Jin, Yuchen Guo, Hongwu Wang, Feng Xu, Guoliang Huang

**Affiliations:** 1Department of Biomedical Engineering, School of Medicine, Tsinghua University, Beijing 100084, China; 2BNRist and School of Software, Tsinghua University, Beijing 100084, China; 3Graduate School, Adamson University, Manila 1000, Philippines; 4Beijing National Research Center for Information Science and Technology, Tsinghua University, Beijing 100084, China; 5Dongzhimen Hospital, Beijing University of Chinese Medicine, Beijing 100700, China; 6Emergency General Hospital, Beijing 100000, China; 7National Engineering Research Center for Beijing Biochip Technology, Beijing 102206, China

**Keywords:** lung neoplasms, sclera image, artificial intelligence (AI), multi-instance learning model

## Abstract

Lung cancer remains the most commonly diagnosed cancer and the leading cause of death from cancer. Recent research shows that the human eye can provide useful information about one’s health status, but few studies have revealed that the eye’s features are associated with the risk of cancer. The aims of this paper are to explore the association between scleral features and lung neoplasms and develop a non-invasive artificial intelligence (AI) method for detecting lung neoplasms based on scleral images. A novel instrument was specially developed to take the reflection-free scleral images. Then, various algorithms and different strategies were applied to find the most effective deep learning algorithm. Ultimately, the detection method based on scleral images and the multi-instance learning (MIL) model was developed to predict benign or malignant lung neoplasms. From March 2017 to January 2019, 3923 subjects were recruited for the experiment. Using the pathological diagnosis of bronchoscopy as the gold standard, 95 participants were enrolled to take scleral image screens, and 950 scleral images were fed to AI analysis. Our non-invasive AI method had an AUC of 0.897 ± 0.041(95% CI), a sensitivity of 0.836 ± 0.048 (95% CI), and a specificity of 0.828 ± 0.095 (95% CI) for distinguishing between benign and malignant lung nodules. This study suggested that scleral features such as blood vessels may be associated with lung cancer, and the non-invasive AI method based on scleral images can assist in lung neoplasm detection. This technique may hold promise for evaluating the risk of lung cancer in an asymptomatic population in areas with a shortage of medical resources and as a cost-effective adjunctive tool for LDCT screening at hospitals.

## 1. Introduction

Lung cancer is the leading cause of death from cancer worldwide (18.0% of the total cancer deaths) [1]. Early detection and treatment can considerably improve 3-year survival rates [2], but over half of patients were diagnosed at an advanced stage [3]. Lung cancer is expected to remain a major public health issue for decades [4].

Low-dose computed tomography (LDCT) has been adopted as a tool for early lung cancer screening for decades. Large-scale LDCT screening has been recommended for people with high risk of lung cancer at major medical and governmental organizations [5,6,7]. Since the first recommendation of lung cancer screening by LDCT in 2013, the United States has seen a steep decline in advanced lung cancer incidence and an increase in 5-year relative survival, as well as a more promising outlook for lung cancer at all stages [2].

As an effective method to distinguish lung cancer at an early stage, it is of the utmost importance to select the population amenable to LDCT screenings [8]. The US Preventive Services Task Force (USPSTF) concludes that age (50 to 80 years old), total cumulative exposure to tobacco smoke (20 packs per year), and years since quitting smoking (15 years) are critical criteria [6]. However, not all lung cancers are caused by active smoking. Background risks resulting from other exposures, such as cooking fumes and environmental pollution, and their interactions between epigenetic and genetic processes also have a great influence on lung cancer occurrence [9]. For example, despite a low smoking prevalence, females in East Asia suffer from a high incidence of lung cancer [10]. Some lung cancer risk-prediction models are also recommended to recognize high-risk individuals [11,12,13] but are deemed unsuitable because they were validated in limited populations or with marginal predictive power [7].

Besides the criteria for selection of the high-risk population, widespread adoption of LDCT screening remains challenging. As large, complicated medical equipment, LDCT screening is out of reach at primary medical institutions, and it requires a relatively complicated operation. Although LDCT can find small nodules on the lung, it cannot distinguish between benign and malignant lung cancer, which causes patient anxiety, unnecessary follow-up, and invasive diagnostic procedures in previously screened patients [14]. Therefore, there is a critical need for a more cost-effective tool for population (pre) screening, such as identifying individuals who are likely to harbor a tumor that can be detected during follow-up LDCT examinations. An adjunctive test to help evaluate the malignancy potential is also urgently needed in order to improve the specificity of noninvasive lung cancer detection and diagnostic triage.

Moreover, the interpretation of images captured by LDCT or other medical examination machines remains challenging [15]. Detection accuracy attained by clinical experts varies widely and leaves room for improvement [16]. Artificial intelligence (AI) is proving to be a capable aid to this challenge. Several studies have proven that AI can meet or outperform human experts on a variety of medical-image analysis tasks [17,18,19,20]. Most recently, deep learning methods for lung cancer prediction have been successful with higher detection accuracy [21,22,23]. Some deep learning systems for lung cancer detection even gain the ability to analyze an entire 3D CT scan, which can provide more diagnostic information about features such as blood vessels [24].

As an internal organ that can be observed externally, the human eye can reveal possible diseases or dysfunctions of specific organs in a painless and non-invasive way [25]. The sclera in the eye, which is visible on the ocular surface, may provide useful information about one’s health status [26]. Neoplasms play a key role in the formation of new blood vessels from pre-existing blood vessels [27]. As the only human blood vessels that can be directly observed without being covered by skin or affected by pigments, dilated scleral vessels have been used to evaluate the health risk of patients who are suspected to have internal carotid artery occlusions [28]. Redness of the sclera may reflect abnormal vasodilation of the conjunctival blood vessels [29]. However, limited work has been reported using scleral images to assist with the screening of populations with high risk of lung diseases [30].

Thus, we hypothesize that lung cancer is related to the blood vessel pattern on the sclera. This study aimed to explore an artificial intelligence system based on scleral images to distinguish benign from malignant lung nodules. Firstly, we developed a novel scleral optics screening instrument to obtain scleral images in a convenient, cost-effective, and non-invasive way. Secondly, we preprocessed scleral images, extracted features, and classified data sets by using deep learning methods so as to find the correlation between lung neoplasms and scleral features. Then, we evaluate the performance of our new AI system for lung cancer detection by using scleral images from patients who have been diagnosed with lung nodules through LDCT screening. Furthermore, we showed how this system might be integrated into screening workflows to speed the advent of population lung cancer screening.

## 2. Materials and Methods

### 2.1. Participants

From March 2017 to January 2019, 3923 participants were recruited through general practitioners at the Emergency General Hospital (Beijing, China). We included adult residents showing pulmonary symptoms, such as coughing, chest pain, and dyspnea, or with at least one high risk factor: (1) current smokers who had at least 20 packs of cigarettes per year or former smokers who had quit smoking no more than 15 years ago; (2) lung cancer history of close family members; (3) long-term exposure to cooking oil fumes (> 50 dish-years); (4) a long history of passive smoking (>2 h/day indoors for more than 10 years). Individuals who had previously been diagnosed with any kind of cancer were excluded from the study. The research protocol was approved by the Ethics Committee of the Emergency General Hospital and Tsinghua University. All enrolled subjects signed a written informed consent form before entering the study.

### 2.2. Screening Strategy

All participants were invited to take an LDCT screening using a 64-detector CT row scanner (Brilliance, Philips, Cambridge, MA, USA). Spiral CT images of the lung were captured with a thickness of 1 mm and were reconstructed with 30% overlap between each layer. Any nonsolid nodules ≥5 mm or solid nodules ≥8 mm were considered positive. A total of 3821 subjects were negative for the LDCT test. Participants with positive LDCT screening findings were asked to take a scleral screening follow-up. The sclera images were taken using a specially developed optical instrument for scleral feature analysis. The scleral images were pre-analyzed to determine whether they qualified for deep learning in the next stage. Seven participants were excluded with invalid scleral images at this stage. For any positive pulmonary nodules, to confirm whether it is benign or malignant, pathological examinations were also conducted by experienced clinicians at the hospital who were blinded to this study. If lung cancer was diagnosed, the participant was split into the malignant group; otherwise, the participant was in the benign group. Finally, we obtained 95 valid subjects with 950 scleral images to enroll in the subsequent AI analysis, including 20 benign subjects and 75 malignant subjects. The AI analysis in this study was performed at Tsinghua University (Beijing, China). Figure 1 shows an overview of the screening strategy.

### 2.3. Scleral Imaging Method and Instrument

In the clinic, ophthalmic examinations are usually performed with slit lamps, which are complicated to operate and time-consuming to post-process. The reason is that because the eyeball has a multilayer quasi-sphere structure, common illumination sources will cause many reflection shadows on the scleral images from the interface of different layers of the eyeball, as shown in the picture on the left side of Figure 2B.

To eliminate the interference caused by reflection shadows and obtain reflection-free scleral images, an adaptive reflection- and shadow-free scleral imaging method was developed, as shown in Figure 2C. S is the illumination source of a 1-W LED white light, G is the cross-guiding light of a 1-W green LED, the lens has a 100-mm focal length, the CCD is a Canon 5D, Ψ is the angle between the optical axis and S, and Ø is the angle between the optical axis and the pupil. To build the neural network-based iris auto-tracking and sclera auto-focusing method, the optimum values of Ψ and Ø were obtained at 40–50° and 65–80°, respectively, when all of the reflection shadows of the illumination source were focused on a small point and superimposed onto the pupil. The optimal values of Ψ and Ø differ for different people. Using the cross-light guide G, the user is guided to rotate the eyeball and adjust the pupil position to give negative feedback on the iris automatic tracking system until the reflection shadows focus on the pupil. The auto-focus sclera can then be clearly imaged without reflection shadows, as shown in the picture on the right side of Figure 2B.

An adaptive reflection- and shadow-free scleral imaging instrument was developed based on the designs mentioned above. Using the indicator light G, subjects were instructed to rotate the eyeball center, up, down, left, and right. Thus, the sclera is synchronously photographed, and images of the entire sclera without the reflection shadows of the illumination source can be obtained within 3 min, as shown in Figure 2A. Scleral images of each subject were considered together as a package of images, which were marked as negative or positive according to the results of the pathological examination, i.e., packages of lung cancer patients were marked as positive, and packages of normal subjects were marked as negative.

### 2.4. Development of AI Models

Since research and open-source datasets about machine learning and deep learning for scleral images are so rare, we built various AI models to explore the association between scleral features and lung neoplasms. Every model consists of target area segmentation, feature extraction, and classification. In our experiment, we first preprocessed images to segment the scleral area, then extracted image features such as blood vessels, followed by a feature classification step, and finally we used models to decide whether the subject was benign or malignant according to the classification result of scleral images.

### 2.5. Statistical Analysis

Diagnostic accuracy, sensitivity, and specificity were utilized to indicate differences between the clinical diagnosis and results by using our scleral screen AI detection system. Accuracy measures the proportion of cases diagnosed correctly to the total number of participants. Sensitivity is the percentage of true positives out of all subjects, and specificity is the proportion of true negatives out of all subjects. Detection accuracy, sensitivity, specificity, and AUC value were calculated using SPSS software (version 18.0), as well as 95% confidence intervals. Means ± SD were used to represent normally distributed data. Expressions of these indices are shown as follows, where *TP* represents true positive, *TN* represents true negative, *FN* represents false negative, and *FP* represents false positive:(1)$$Accyracy=\frac{TP+TN}{TP+TN+FN+TN},$$
(2)$$Sensitivity=\frac{TP}{TP+FN},$$
(3)$$Specificity=\frac{TN}{TN+FP}.$$

## 3. Results

### 3.1. Characteristics of Subjects Enrolled in AI Analysis

Of the 95 subjects enrolled that were prospectively eligible for AI analysis, the main characteristics were presented according to the study group in Table 1. At the time of scleral screening, the 75 subjects diagnosed with malignant lung nodules were significantly older (average 61.9 years old) than those in the benign group (average 50.6 years old). A total of 33 females represented 34.7% of the enrolled AI analysis sample, and 83.9% of them were in the malignant group, while 68.7% of male subjects were diagnosed as having lung cancer. For the malignant subjects, the majority type of lung cancer was lung squamous cell carcinoma (LUSC), with a percentage of 37.3%, while the least prevalent was small cell lung cancer (8.0%). The proportions of lung metastasis, lung adenocarcinoma (LUAD), and mixed/unspecified NSCLC were 22.7%, 20.0%, and 12.0%, respectively.

### 3.2. Modeling of AI Models

Figure 3 shows the structure of the three best-performing models, and we combined them as the backbone algorithm of our non-invasion AI method.

In the first model, the collected scleral images were preprocessed before being fed. Regions of the sclera were annotated by a bounding box on each image and then cropped and resized as a separate image of size 512 × 384 px. For each subject, ten scleral images were captured, five for the left eye and the other five for the right eye. Images of the right eye were flipped horizontally to ease the deep learning model.

To improve the model’s performance, we used the technique of transfer learning, which first pre-trains a model on data from a source domain and then transfers the weights to further train on data from a target domain. The insight in transfer learning is that the source and target domains share some basic feature patterns. In our experiment, we first pre-trained ResNet-18 on ImageNet, a large image database, then fine-tuned the model on scleral images. We performed data augmentation by jittering the brightness, saturation, and contrast of the input images. To overcome the problem of class imbalance, we optimized the MIL model with focal loss63 as the target function. The model was optimized by Adam optimizer64 with an initial learning rate of 0.0001.

Then, we fed all the images of each subject into the Resnet-18 network, and a sequence of feature vectors was extracted. For each patient, different regions of the two eyes were revealed in different images. We performed data augmentation by jittering the brightness, saturation, and contrast of the input images. To overcome the problem of class imbalance, we optimized the multi-instance learning (MIL) model with focal loss as the target function. The MIL model was applied to all images of the patient to make a decision rather than predicting from single images. Using the MIL model, a fusion feature vector $h_{fusion}=\frac{1}{N}\sum_{i=1}^{N}h_{i}$ is aggregated from ${\left\{h_{i}\right\}}_{i=1}^{10}$ using average pooling. Finally, $h_{fusion}$ is fed into a multi-layer perceptron model (MLP) to obtain a prediction score, as shown in Figure 3A.

In the second model, we used U-net to segment the sclera areas, then used the autoencoder to extract the features of the image, which is the backbone of the whole model, and finally used the support vector machine for classification. The contracting path and expanding path are 29 layers in total, as shown in Figure 3B. It is trained from scratch, and we used data augmentation such as flipping and rotation to optimize the training effect. The overall structure of the third model is similar to the second one, except the target area segmentation part is changed into a crop segmentation algorithm and the feature extraction part is changed into Vgg16-net merely, as shown in Figure 3C.

### 3.3. Performance of the Top Three AI Models

After fine tuning, the optimal point performance of the different algorithms mentioned above is shown in Table 2. The model based on Resnet-18 and MIL performs best overall; the accuracy is up to 0.811, which is over 4% higher than the second place and nearly 6% higher than the third one. Thus, we used the first model for the further testing of different input strategies.

### 3.4. Comparison of Different Scleral Image Input Strategies

Further, we compared different scleral image input strategies based on the No.1 model; all other experimental settings were kept the same as previous experiments. Scleral image input strategies, including all ten images, treated the left and right eyes of each subject as two separate samples and eight scleral images other than the center ones. The binary classification results of the four input strategies were averaged across all of the test sets of the thrice-repeated three-fold cross-validation procedures. We utilize accuracy, sensitivity, specificity, AUC, as shown in Table 3, and the ROC curve (see Figure 4) to assess the performance.

After fine tuning, the eight scleral images input strategy achieved a mean AUC of 0.897 (95% CI = 0.856–0.938). At the optimal point, average accuracy, sensitivity, and specificity were 0.835 ± 0.031, 0.836 ± 0.048, and 0.828 ± 0.095, respectively. The MIL model that treated the two eyes of each subject as two separate samples achieved a mean AUC of 0.864 (only left eye) and 0.850 (only right eye), which is nearly 4% lower than our method, while the MIL models using all ten images achieved a mean AUC of 0.867, which is 3% lower than our method. That is probably because the two central images do not show much scleral information and are different from the other images. The above experiments show that using eight scleral images other than the center yielded the best performance with a high AUC of 0.897.

## 4. Discussion

Clinically, chest X-rays remain the most common and least expensive imaging modality to detect pulmonary abnormalities related to the early stages of the disease. Computerized tomography (CT) scans remain the standard of care for further evaluation of suspected lung cancer. However, X-ray and CT can only detect tumors of ~1mm or larger in size, and they cannot distinguish tumors from lung infections, indicating low sensitivity and high false-positive rates. Magnetic resonance imaging (MRI) and positron emission computerized tomography (PET) have also been deployed for the diagnosis of lung cancer, but they are too expensive and out of reach at primary medical sites. Meanwhile, large-scale imaging equipment with a danger of radiation exposure must be used to perform CT scans, which results in high costs. Biomarkers can also be used for the early detection of lung cancer, but only trace levels of biomarkers exist in the cancerous cells at an early stage, so the accuracy of cytology is poor.

Radiologists already employ computer-aided diagnostic tools to assist them in detecting malignant tumors, basically to tell the system to look for features related to malignancies. Additionally, researchers are attempting to develop methods for lung cancer diagnosis more efficiently and quickly. Among them are methods to make lung-cancer screening more precise and accessible to all, especially for people in areas that suffer from a shortage of large-scale equipment, such as a CT imaging system. However, there is a long way to go before new systems become clinical mainstays because a careful nurturing of the relationship between features of data and the machines on which they depend is needed.

From a point-of-care perspective, our AI method based on a non-invasive scleral screen instrument has various advantages, such as simplicity, being remotely available, low cost, requiring no reagent or a specialized laboratory setting, and being almost real-time (<3 min), making it suitable for large-scale electronic health monitoring. More importantly, due to its painless and non-invasive nature, we believe our assessment system for lung cancer at an early stage may have the chance to simplify physical examination processes and improve patients’ medical experiences.

Many aspects are worth exploring further in our research. First, the participants in our study all had lung symptoms or high-risk factors for lung cancer, so we obtain a higher rate of malignant subjects than benign subjects after LDCT screening, which causes an imbalance of data in the building of AI models. We will include more healthy subjects in our later study. Second, we only recruited participants at one hospital, which may limit the mobility of the algorithm. Thus, further external multi-center validation is warranted. Third, the non-invasive AI method could be used to detect malignant lesions in the lungs, but whether it could distinguish subtypes requires more samples from patients with different subtypes of lung cancer. Furthermore, even though AI applied to biomedical research is a powerful tool to analyze deep features and connections among lung cancer and scleral images, privacy protection and improper use with social implications are concerns.

## 5. Conclusions

In conclusion, we developed an AI method based on a non-invasive scleral screen instrument for predicting the risk of lung cancer. We developed an adaptive reflection- and shadow-free scleral imaging instrument to capture scleral images in a non-invasive way, which could conveniently and quickly collect complete scleral images in four directions and perform AI analysis in 3 min without any reagent consumption or the need for a laboratory. We also developed a multi-instance learning model to distinguish benign from malignant lung nodules by using scleral images. The binary classification results of the MIL model achieved an average AUC of 0.897, which indicates great potential for early screening of practical lung cancer during periodic physical checkups or daily family health monitoring.

Our results proposed a new concept: that the analysis of scleral images using deep learning can help detect lung cancer in this innovative study. This effort supported a potential step towards the development of a deep learning-based tool for the pre-test lung cancer probabilities assessment in outpatient clinics or through telemedicine in the community, which may help to evaluate the risk of lung cancer in an asymptomatic population in areas with a shortage of medical resources or as a cost-effective adjunctive tool for LDCT screening at hospitals.

## Figures and Tables

**Figure 1 diagnostics-13-00648-f001:**
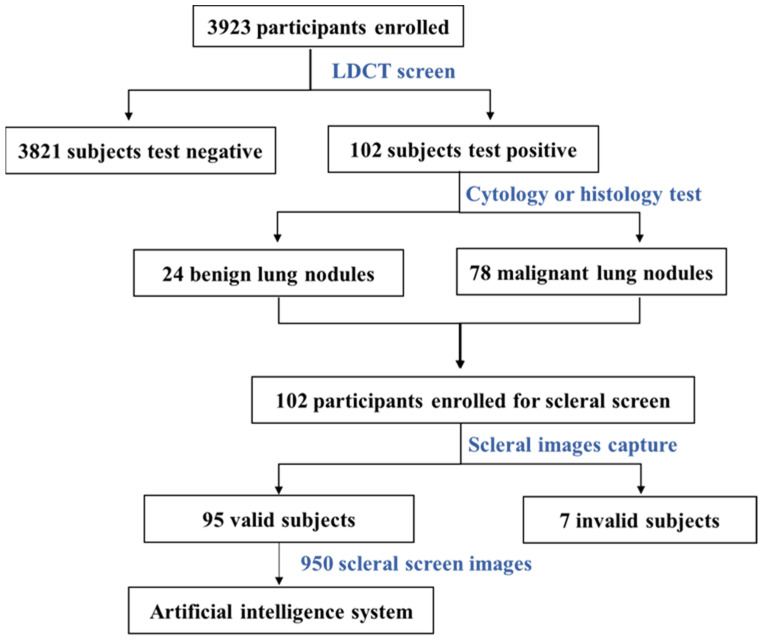
Flow chart illustrating inclusion and exclusion criteria applied to screening and AI analysis participants.

**Figure 2 diagnostics-13-00648-f002:**
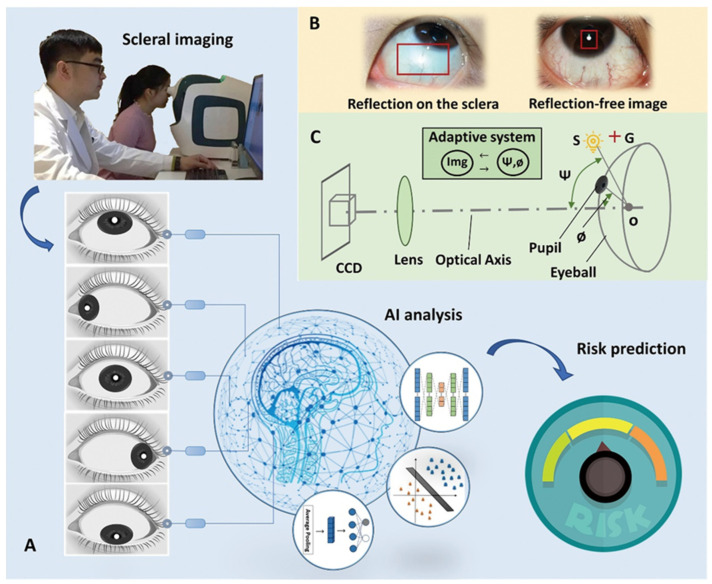
Reflection- and shadow-free scleral screen instrument and workflow of image analysis. (**A**) Workflow of the non-invasive AI method for detecting lung neoplasms based on scleral images. (**B**) Imaging the sclera directly (**left**) and using the novel scleral imaging instrument (**right**). (**C**) Schematic of the novel scleral imaging instrument.

**Figure 3 diagnostics-13-00648-f003:**
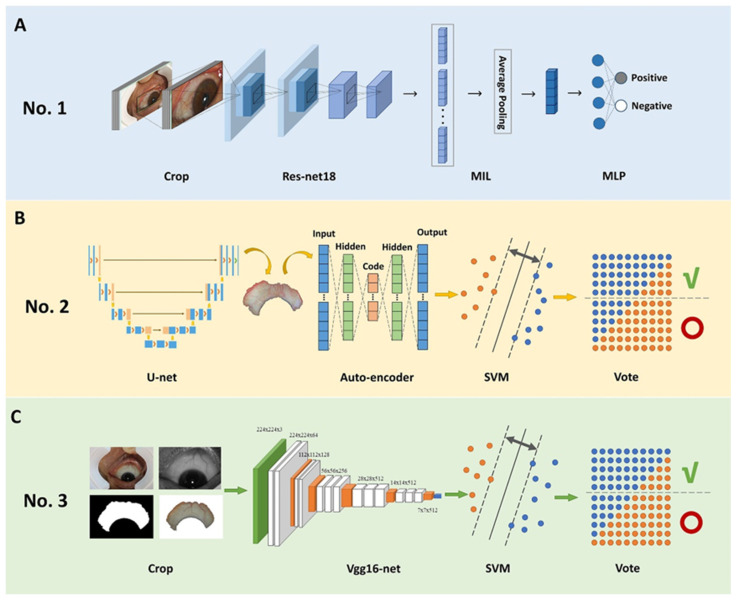
Structure of the three best performing AI models. (**A**) The first model developed uses a bounding box to annotate the region of the sclera, Resnet-18 to extract features, and MIL and MLP to classify. (**B**) The second model uses U-net to segment the scleral area, autoencoder to extract the features, and SVM and vote to classify. (**C**) The third model uses a traditional threshold algorithm to segment the scleral area, Vgg16-net to extract features, and SVM and vote to classify.

**Figure 4 diagnostics-13-00648-f004:**
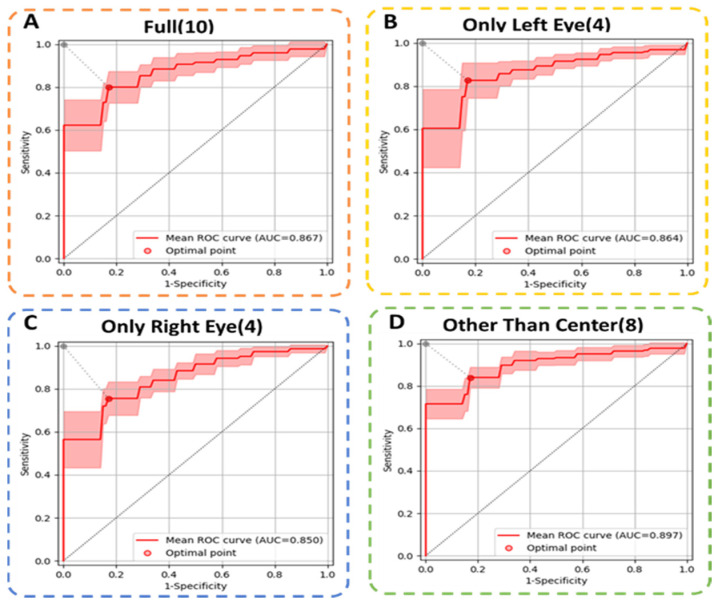
ROC curves of different scleral image input strategies. (**A**) The mean ROC curve with 95% CI and the optimal point for ten images; (**B**) The mean ROC curve with 95% CI and the optimal point for only the left eye (four images); (**C**) The mean ROC curve with 95% CI and the optimal point for only the right eye (four images); (**D**) The mean ROC curve with 95% CI and the optimal point for images other than the center (eight images). The dashed boxes in different colors represent different input strategies.

**Table 1 diagnostics-13-00648-t001:** Main characteristics of subjects enrolled in AI analysis.

Characteristics	Benign Group	Malignant Group
Age	50.6	61.9
Gender		
Female	10 (30.3%)	23 (68.7%)
Male	10 (16.1%)	52 (83.9%)
Tumor type		
Lung squamous cell carcinoma (LUSC)		28 (37.3%)
Lung metastasis		17 (22.7%)
Lung adenocarcinoma (LUAD)		15 (20.0%)
Mixed/unspecified NSCLC		9 (12.0%)
Small Cell Lung Cancer (SCLC)		6 (8.0%)

**Table 2 diagnostics-13-00648-t002:** Performance of the top three AI models.

Models ^1^	Accuracy	Sensitivity	Specificity
No. 1	0.811	0.813	0.800
No. 2	0.779	0.827	0.600
No. 3	0.768	0.827	0.550

^1^ Model No. 1 uses a bounding box to annotate the region of sclera, Resnet-18 to extract features, MIL and MLP to classify; Model No. 2 uses U-net to segment scleral area, auto-encoder to extract features, SVM and vote to classify; and Model No. 3 uses a crop algorithm to segment scleral area, Vgg16-net to extract features, SVM and vote to classify.

**Table 3 diagnostics-13-00648-t003:** Results of comparisons between different scleral image input strategies.

Input Images ^2^	Accuracy	Sensitivity	Specificity	Average AUC
Full (10)	0.818 ± 0.043	0.818 ± 0.044	0.817 ± 0.090	0.867 ± 0.058
Only Left Eye (4)	0.835 ± 0.044	0.849 ± 0.054	0.786 ± 0.084	0.864 ± 0.063
Only Right Eye (4)	0.779 ± 0.055	0.778 ± 0.061	0.783 ± 0.051	0.850 ± 0.055
Other Than Center (8)	0.835 ± 0.031	0.836 ± 0.048	0.828 ± 0.095	0.897 ± 0.041

^2^ Including full eyes (ten images), only left eye (four images), only right eye (four images), and other than center (eight images), in which ± indicates the 95% CI.

## Data Availability

Download links of the original scleral images: https://cloud.tsinghua.edu.cn/f/daf5509e53fc48c6b21f/?dl=1 (accessed on 2 January 2023) or https://drive.google.com/file/d/1hQOvS8pclx4GsjVuATkTHWGEQjWcYSpD/view?usp=sharing (accessed on 29 January 2023).

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
