# Peer review of "Machine Learning System for Lung Neoplasms Distinguished Based on Scleral Data"

_diagnostics, 2023, doi:10.3390/diagnostics13040648_

Round 1
Reviewer 1 Report
This research report very important results on the fact that scleral features such as blood vessels may associate with lung cancer and the non-invasive AI method based on scleral images can assist in lung neoplasm detection. The proposed solution was evaluated on large dataset consisted on 3923 subjects and obtained satisfying results.
The dataset is available to download however please prepare the link to not be partitioned between lines because I succeeded in making a correct url after several tries.
"The collected scleral images were preprocessed before being fed into the classification model." - there are no details about the classification model besides the fact that a transfer learning was used for training. Please describe your method in details.
Later in text there is information that the ResNet-18 pretrained on ImageNet was used however there are no details about the classification procedure. Please supply that information.
The second algorithm uses U-net for segmentation however it is unclear to me how the segmented region is than utilized. Please make the precise description of U-Net model (for example what is a backbone of the model, how many layers it has, is it trained from scratch etc.). The descriptive image might be helpful.
Reviewer 2 Report
Please find my comments on the pdf file

Round 2
Reviewer 1 Report
Authors have addressed my remarks. In my opinion paper can be accept in present form.